# Pixel Diffuser: Practical Interactive Medical Image Segmentation without Ground Truth

**DOI:** 10.3390/bioengineering10111280

**Published:** 2023-11-02

**Authors:** Mingeon Ju, Jaewoo Yang, Jaeyoung Lee, Moonhyun Lee, Junyung Ji, Younghoon Kim

**Affiliations:** 1Major in Bio Artificial Intelligence, Department of Applied Artificial Intelligence, Hanyang University at Ansan, Ansan 15588, Republic of Korea; msgee@hanyang.ac.kr (M.J.); onnoo@hanyang.ac.kr (J.Y.); wayexists02@gmail.com (J.L.); jee9894@hanyang.ac.kr (J.J.); 2Major in Bio Artificial Intelligence, Department of Computer Science & Engineering, Hanyang University at Ansan, Ansan 15588, Republic of Korea; greenzip0510@hanyang.ac.kr

**Keywords:** interactive medical segmentation, iterative segmentation, CT segmentation, autoencoder, reconstruction noise

## Abstract

Medical image segmentation is essential for doctors to diagnose diseases and manage patient status. While deep learning has demonstrated potential in addressing segmentation challenges within the medical domain, obtaining a substantial amount of data with accurate ground truth for training high-performance segmentation models is both time-consuming and demands careful attention. While interactive segmentation methods can reduce the costs of acquiring segmentation labels for training supervised models, they often still necessitate considerable amounts of ground truth data. Moreover, achieving precise segmentation during the refinement phase results in increased interactions. In this work, we propose an interactive medical segmentation method called PixelDiffuser that requires no medical segmentation ground truth data and only a few clicks to obtain high-quality segmentation using a VGG19-based autoencoder. As the name suggests, PixelDiffuser starts with a small area upon the initial click and gradually detects the target segmentation region. Specifically, we segment the image by creating a distortion in the image and repeating it during the process of encoding and decoding the image through an autoencoder. Consequently, PixelDiffuser enables the user to click a part of the organ they wish to segment, allowing the segmented region to expand to nearby areas with pixel values similar to the chosen organ. To evaluate the performance of PixelDiffuser, we employed the dice score, based on the number of clicks, to compare the ground truth image with the inferred segment. For validation of our method’s performance, we leveraged the BTCV dataset, containing CT images of various organs, and the CHAOS dataset, which encompasses both CT and MRI images of the liver, kidneys and spleen. Our proposed model is an efficient and effective tool for medical image segmentation, achieving competitive performance compared to previous work in less than five clicks and with very low memory consumption without additional training.

## 1. Introduction

Medical image segmentation has been a crucial task for identifying structures of specific organs. In recent years, deep learning has emerged as a promising approach for solving medical segmentation problems [1,2,3,4,5]. Despite the excellent performance of deep-learning-based models, training them requires a significant amount of data with ground truth labels, which can be time-consuming and resource-intensive.

Interactive medical segmentation can alleviate these challenges by helping us easily obtain segmentation labels. Unlike traditional image segmentation, interactive medical segmentation takes various user interactions (e.g., bounding boxes [6], scribbles [7] or clicks [8]) to compute the region of segmentation, leading to a substantial reduction in labeling cost. However, existing works [6,7,8] still require the use of segmentation labels during training.

Recently, general interactive segmentation models such as RITM [9], SimpleClick [10], and SAM [11] have shown to be effective even for medical image segmentation. These models generate and refine segmentation predictions using user clicks or points provided by the user. Despite being trained with general image segmentation data, they can also be applied to medical image segmentation without requiring additional training on medical images. However, alongside their impressive results in general and medical image segmentation, we observed that they tend to capture much wider regions than the ground truth masks, which leads to an increased number of clicks. We analyzed that this is due to the fact that they are attempting to predict the segmentation mask in a single step.

This study proposes an interactive medical segmentation model named PixelDiffuser, which consists of a light weight pre-trained autoencoder trained on general image data. Using the proposed method, a user can obtain a segmentation mask by simply clicking the location of interest. When the click is given, our model first transforms it into a binary circle image centered at the click location with a 1-pixel radius. Subsequently, the circle image is gradually transformed into the segmentation mask, as the name suggests.

Our segmentation procedure is based on our key observation, reconstruction noises, which are distortions that arise in decoded images generated through the encoding and decoding process of an autoencoder. The most important aspect is that if we mix two images in the encoding space (feature space) and decode the mixed latent, the resulting image appears as an intermediate between two source images. Motivated by this observation, our approach iteratively mixes the click image and the medical image to make the click image cover the organ of interest.

Since our method utilizes a simple autoencoder, it does not require any medical or general segmentation data, which is converse to the previous works, such as SimpleClick, SAM, and MedSAM which rely on either medical or general image segmentation data.

We employed The Beyond The Crutial Vault (BTCV) dataset [12], a multi-organ abdominal CT reference, and CHAOS dataset [13], which aims at the segmentation of liver, kidneys and spleen from CT and MRI data, to validate the model. For evaluation, we utilized the Dice score, comparing our model’s performance to other interactive segmentation approaches. It is important to note that MedSAM [14], which specializes in medical image segmentation based on the SAM method, is trained on large-scale medical image segmentation data. Therefore, it differs from our approach, which is an interactive medical segmentation model that does not require training using medical image segmentation data. For this reason, we exclude MedSAM [14] from the evaluation. The experimental results indicate that PixelDiffuser generates more accurate segmentation masks with only a few clicks. In addition, our method achieves comparable results to the SAM Huge model while being 91.4× lighter than SAM-Huge.

Our contributions can be listed as follows:In this work, we present an innovative interactive medical segmentation model called PixelDiffuser. Unlike traditional approaches, our model does not necessitate specialized training on medical segmentation datasets. Instead, we leverage a simple autoencoder trained on general image data.The PixelDiffuser operates interactively by initiating the segmentation process from a single click point. This strategic approach ensures that the model gradually identifies the target mask, thereby mitigating the risk of overstepping the actual ground truth region. Consequently, the need for excessive user inputs is significantly reduced.Through comprehensive experiments, we showcase the remarkable efficiency of our proposed algorithm. With only a minimal number of clicks, our approach outperforms alternative methods in achieving accurate target segmentation.

## 2. Related Works

### 2.1. Medical Image Segmentation

Medical image segmentation is an important task in assisting diagnosis and surgeries. As convolutional neural networks [15] have shown high performance, many other works utilized them to solve medical image segmentation. U-Net [5] is inspired by [15] and achieved state-of-the-art in medical image segmentation. After that, many variants of U-Net [1,2,3,4] have been developed. TransUNet [1] and UNETR [3] employed transformer [16] as a feature extractor, before predicting segmentation maps. Swin-UNet [2] and Swin-UNETR utilized Swin-Transformer architecture [17] to acquire contextual features from the medical images.

Additionally, to improve the performance of medical segmentation, ensemble-based methods [18,19,20,21] have been also invented. DivergentNets [18] proposed TriUnet which consists of three separate U-Net, and combined it with an ensemble of previous state-of-the-art medical segmentation models. This ensemble, referred to as DivergentNets, has an advantage in improving generalization in medical image segmentation. ALMN (Affinity Learning-based Multi-branch Ensemble Network) [19] employed an affinity-learning-based ensemble learning framework to achieve high performance in 3D medical segmentation. Dang, Truong et al. [20] introduced a combining strategy of several segmentation models. Georgescu, Mariana-Iuliana et al. [21] leveraged the diversity of several segmentation models to generate an efficient ensemble combining strategy. However, the methods discussed earlier require ground truth labels for training the segmentation model, which comes with the drawback of demanding substantial effort and time to collect.

### 2.2. Medical Interactive Image Segmentation

Traditional interactive segmentation methods can be categorized depending on the type of user interaction, as follows: contour-based methods [22], bounding-box-based methods [23], and scribble-based methods [24,25,26,27]. Recently, deep-learning-based methods have improved performance with fewer user interactions. BIFSeg [6] takes a bounding box from a user to enable zero-shot segmentation. DeepIGeoS [7] requires a scribble-type user interaction to find segmentation predictions of fetal MRI or 3D brain tumor. MIDeepSeg [8] performed 2D and 3D medical image segmentation through user-provided interior margin points. Moreover, there has been research that utilizes models trained with non-medical image data for interactive medical image segmentation. For instance, models like SimpleClick [10] and SAM [11] demonstrate the capability of segmentation even for medical images despite being trained with general image segmentation data.

## 3. Motivation

In contrast to the previous methods that predict a full segmentation mask in a single step, our approach predicts the final segmentation mask by iteratively expanding from the initial point. This method is inspired by our key observation named reconstruction noises, distortions in reconstructed images stemming from the bottleneck of the autoencoder architecture. In this section, we introduce the reconstruction noises and describe how we utilize them to compute the segmentation mask.

### 3.1. The Reconstruction Noises

An autoencoder architecture comprises two primary components: an encoder (*E*) and a decoder (*D*). For a given grayscale image *a*, the encoder calculates a latent feature $z_{a}=E\left(a\right)$, and the decoder reconstructs the image as $\tilde{a}=D\left(E\left(a\right)\right)$. Due to the bottleneck in autoencoders, there is an information loss which inevitably leads to distortions in the reconstructed image.

This distortion effect is even more evident when mixing two images in the feature space. Consider another grayscale image *b* with its latent feature $z_{b}=E\left(b\right)$. By averaging $z_{a}$ and $z_{b}$, we obtain an averaged latent feature $z_{m}$. When decoded, this produces a blended image *m*, which is a mix of *a* and *b*. However, due to the information loss stemming from a bottleneck, *m* does not perfectly align with a direct pixel-level mix of *a* and *b*. As illustrated in Figure 1, mixing at deeper layers introduces more distortions. We term these distortions as reconstruction noises.

### 3.2. From the Reconstruction Noises to Segmentation Mask

Based on our observation, we introduce our intuition on how the reconstruction noises can be utilized to generate segmentation masks. As depicted in Figure 1, it becomes evident that when the features of two images are mixed at deeper layers and then reconstructed, the resulting shape appears to be an intermediate between the two source images, *a* and *b*. This blending effect of the images is more pronounced in deeper layers. Drawing on this intuition, we posed the following question: If we are provided with a CT image and a click image (the dot image), could we potentially obtain segmentation masks through their iterative mixing? This is our motivation for the proposed method, PixelDiffuser, which generates segmentation masks by iteratively mixing a CT image with a click image (the dot image). We describe the method in more detail in the following section.

## 4. The PixelDiffuser

We propose the PixelDiffuser, an interactive medical segmentation method. Given a CT image I and a user’s click, our model produces an initial segmentation by the initial segmentation process. This iterative procedure transforms the user’s click into a segmentation. Within this process, the click is represented as a circle image, $g_{0}$, centered at the click location with a 1-pixel radius. This circle image is then gradually expanded into the initial segmentation mask, $g_{T}$, by iteratively blending with the CT image, I. If the initial segmentation is not satisfied, additional clicks can be provided to refine the segmentation using refinement process, which adds the additional region to $g_{T}$ or removes the irrelevant region from the $g_{T}$. Each step of this process is described in the following subsections.

Motivated by reconstruction noises, our method computes the segmentation mask by iteratively mixing the CT image and the dot image in the feature space to distort the dot image. Our method utilizes the VGG19-based encoder *E* and the decoder *D* [28] to encode and mix the images. The encoder is the convolutional part of VGG19, while the decoder is a neural network symmetrically shaped to the encoder. The autoencoder is pre-trained using the general images (e.g., COCO dataset [29]), and we do not tune it with any medical images.

### 4.1. Initial Segmentation Process

Given the CT image, I, and the dot image, $g_{0}$, created based on the user’s click, the initial segmentation process calculates the initial segmentation mask, $g_{T}$. This is achieved by iteratively blending the CT image and the dot image in the feature space. Figure 2 illustrates an overview of this initial segmentation process. Since the procedure is iterative, we detail the steps involved in computing the intermediate segmentation mask, $g_{t+1}$, given the CT image, I, and the preceding mask, $g_{t}$.

With the two images, I and $g_{t}$, the singular step for computing the intermediate mask, $g_{t+1}$, consists of three sub-steps: (1) density-based intensity adjustment (Section 4.1.1), (2) feature mixing (Section 4.1.2) and (3) segmentation (Section 4.1.3). Density-based intensity adjustment step adjusts the pixel values of I to suppress pixels from irrelevant regions. As a result, reconstruction noises spread to the only target region. In the feature mixing step, both images are blended in the feature space, producing a combined image that carries reconstruction noises. In the segmentation step, the merged image is binarized using the Otsu threshold method [30], yielding the result, $g_{t+1}$.

#### 4.1.1. Density-Based Intensity Adjustment

Consider the appearance of a CT image. Within such an image, numerous white regions, each representing different organs, coexist alongside the target region, which represents the specific organ of interest. This implies that when distorting the dot image, it can be mixed with the other organs. To avoid this, we first need to transform the intensities of the CT image, I, to minimize the influence of pixels from irrelevant regions. The density-based intensity adjustment transforms the CT image, I, to the intensity-adjusted image, x. Let $F_{I}:\left(I,g_{t}\right)\to x$ represent the density-based intensity adjustment function. This function accepts two images, I and $g_{t}$, and outputs the transformed image, x. The definition of $F_{I}$ is as follows:(1)$$\begin{array}{c} F_{I}\left(I,g_{t}\right)=\mathrm{kde}\left(\mathrm{crop}\left(I,g_{t}\right)\right)\left(I\right), \end{array}$$
where crop denotes the cropping operator extracting regions from I masked by $g_{t}$ as shown in Figure 3. The size of this cropped region is determined after the dilation of $g_{t}$ for 7 iterations using a 3×3 kernel, resulting in an effective padding of approximately 14 pixels on each boundary: top, bottom, left, and right. Additionally, kde(i) is a kernel density estimator function [31] which computes the density of pixel values of an image, *i*. Specifically, kde(i)(·) is a function that receives pixel values as input and returns the corresponding density value for those pixels. As illustrated in Figure 3, upon the execution of kde, all pixel intensities are substituted with normalized pixel histogram values. Subsequently, as depicted in Figure 4, given that pixel intensities in the region corresponding to the organ of interest show the highest density, we conduct a density-based intensity adjustment, retaining only those pixels similar to the organ with elevated values.

#### 4.1.2. Feature Mixing

Given the intensity-adjusted image, x, and the intermediate mask, $g_{t}$, the feature mixing step encodes them into the latent features, and subsequently, averages those features to produce the mixed features $f_{a(x,g_{t})}$.
(2)$$\begin{array}{cc} f_{x} & =E(x), \end{array}$$
(3)$$\begin{array}{cc} f_{g_{t}} & =E(g_{t}), \end{array}$$
(4)$$\begin{array}{cc} f_{a(x,g_{t})} & =\frac{f_{x}+f_{g_{t}}}{2}, \end{array}$$
where $f_{x}$ and $f_{g_{t}}$ represent the latent features of x and $g_{t}$, respectively. Using this mixed latent feature, we decode it to obtain the mixed and distorted image, $a\left(x,g_{t}\right)=D\left(f_{a(x,g_{t})}\right)$, as illustrated in Figure 2.

#### 4.1.3. Segmentation

While the density-based intensity adjustment step eliminates many irrelevant regions, the mixed image, $a(x,g_{t})$, still contains regions outside the target organ. To remove the irrelevant regions, we compute another mixed image, a(x,o), where o is a completely black image.
(5)$$\begin{array}{cc} f_{x} & =E(x), \end{array}$$
(6)$$\begin{array}{cc} f_{o} & =E(o), \end{array}$$
(7)$$\begin{array}{cc} f_{a(x,o)} & =\frac{f_{x}+f_{o}}{2}, \end{array}$$
(8)$$\begin{array}{cc} a(x,o) & =D(f_{a(x,o)}). \end{array}$$

We then compute the difference between a(x,o) and $a(x,g_{t})$ to eliminate irrelevant regions. Following this, the Otsu thresholding method [30] is employed to extract the binary segmentation mask $g_{t+1}$. The Otsu method determines an optimal threshold by differentiating two mixed distributions, making it suitable for dividing light and dark regions in the image. During the initial iterations, the computed threshold might exhibit instability, leading to variations in the segmented region. This iteration continues until $g_{t}$ and $g_{t+1}$ are identical. Once the iterative change has stabilized or converged, the process is terminated, and the resulting segment is produced.

### 4.2. Refinement Process

Once the segmentation $g_{t}$ is obtained, the user has the option to provide additional clicks, either positive or negative, for revisions. If $g_{t}$ covers a smaller area than expected, users can offer positive clicks to expand the region. Conversely, if the segmentation mask contains irrelevant areas, negative clicks can be used to eliminate them. Upon receiving a positive click, the PixelDiffuser produces another segmentation mask using the same method as described in Section 4.1 and then combines it with $g_{t}$.

While handling positive clicks is simple, managing negative clicks is a little bit more complex. Upon receiving a negative click, a methodology similar to the initial segmentation process (Section 4.1) is employed to identify the negative region within the positive region. As depicted in the rightmost image of Figure 5, both the positive and negative densities are computed based on their respective click points. The standard deviation of a designated region, whether positive or negative, is determined by the standard deviation of the distances between the points in that region and its corresponding click point (either positive or negative). Given these densities, any region where the negative density surpasses the positive density is removed from the segmentation mask, $g_{t}$.

## 5. Experiments

### 5.1. Dataset

#### 5.1.1. BTCV

We utilize The Beyond The Crutial Vault (BTCV) [12] dataset, which contains the CT volumes for 13 organs—(1) spleen, (2) right kidney, (3) left kidney, (4) gallbladder, (5) esophagus, (6) liver, (7) stomach, (8) aorta, (9) inferior vena cava, (10) portal vein and splenic vein, (11) pancreas, (12) right adrenal gland and (13) left adrenal gland. For each organ, there are 864, 897, 905, 359, 689, 1542, 1015, 2042, 1884, 545, 746, 319 and 369 images, respectively.

For preprocessing, all CT images are resized to 512 by 512 pixels, except in the case of the SimpleClick experiments where a size of 448 by 448 was used. Additionally, pixel values in all images were clipped to a range of [−135, 215] and normalized to [0, 1].

#### 5.1.2. CHAOS

We also evaluate our model’s performance using CHAOS [13] dataset. This is an abdominal medical segmentation dataset, including CT and MRI images of 40 different patients. While the CT dataset only includes segmentation masks for the liver, the MRI dataset contains segmentation masks for four organs—the liver, left and right kidney, and spleen.

For preprocessing of CT images, we resize all images to 512 by 512. Additionally, we clip the pixel values to a range of [−135, 215], which is the same as BTCV. For the MRI dataset, all images are clipped to a range of [100, 1000]. After clipping, we normalized the pixel values to [0, 1].

### 5.2. User Interaction

To conduct a quantitative evaluation, we simulated user clicks using [9] which provides a simulation algorithm to sample both positive clicks and negative clicks. During the segmentation process, it starts with a single positive click and utilizes both negative and positive clicks for revision. It is important to note that while the sampling process requires the true segmentation masks, our proposed model does not rely on these masks during training or real-world applications.

### 5.3. Implementation Details

To implement our approach, we employ a VGG-based encoder *E* and a decoder *D*, where the encoder and decoder are pre-trained on the COCO dataset [29]. Instead of pre-training them from scratch, we adopt pretrained weights from Texture Reformer [32] implementation. The encoder consists of the convolutional components of VGG19 [28], but we only utilize 3 out of the 5 convolutional blocks from VGG19, where a convolutional block represents a sequence of convolutional layers followed by one max pooling. We empirically chose to use only 3 blocks for the encoder, as deeper encoders with more than 4 blocks tend to introduce more distortion. The decoder’s architecture mirrors the encoder’s but uses the nearest upsampling layer instead of max pooling. Just like the encoder, the decoder is implemented using only 3 convolutional blocks. The detailed architectures of them are indicated in Table 1.

Before feeding CT images into the segmentation process, all images are resized to a resolution of 512×512. Additionally, we clip pixel values outside of the range of [−135,215] to adjust intensities. Subsequently, we normalize the images to have pixel values in the range of [0,1].

All experimental evaluations were conducted on a Linux workstation running Ubuntu 20.04, equipped with 16 GB of RAM, an Intel i9 CPU, and an NVIDIA GTX 1080 Ti GPU. The PyTorch 1.12 [33] framework was used for deep learning tasks. Since we employed pretrained weights for the encoder and decoder, we did not implement any code relevant to training, such as an optimizer or training iterations.

### 5.4. Description of Other Methods

We compare our approach with two existing methods, RITM [9], SimpleClick [10] and SAM [11]. Prior to proceeding to the evaluation, we briefly present the description of them. We exclude MIDeepSeg [8], DeepIGeoS [7], and MedSAM [14], which require careful clicks, such as clicks on boundary lines, or rely on medical segmentation data since we are focusing on developing interactive segmentation method when no medical segmentation data for training is available for various reasons.

#### 5.4.1. RITM

This approach [9] is a click-based interactive segmentation model that iteratively learns from user clicks using a CNN-based backbone structure. It receives positive and negative clicks from the user, scales them to match the first layer of the model, and processes them element-wise before inputting them into the model. RITM then makes external masks, corrects the segmentation process, and repeats this procedure.

#### 5.4.2. SimpleClick

SimpleClick [10] is a transformer-based general interactive segmentation model that takes an image and a click as an input. SimpleClick exploits the ViT as a backbone network to extract features from the concatenation of the image and the click (dot image). Utilizing the features, the simple feature pyramid [10] and subsequent segmentation head decode the feature into a segmentation mask.

#### 5.4.3. SAM

SAM, denoted as Segment Anything Model [11], accepts various inputs such as points, boxes, and text in addition to images. Each piece of information is processed by the encoder, and the decoder subsequently produces the segmentation mask. This architecture facilitates segmentation tasks for a multitude of prompts without necessitating further training. Specifically, to achieve zero-shot performance, it undergoes training on over 1 billion mask datasets named SA-1B repeatedly.

#### 5.4.4. FastSAM

As the name suggests, FastSAM [34] enhances the inference speed of SAM by substituting the Vision Transformer [35] with YOLOv8-seg [36] for the backbone network. This change greatly reduces the encoding time for images, resulting in a much faster inference speed compared to SAM. Similar to SAM, it also supports both unconditional and conditional segmentation, using either bounding boxes or click points.

### 5.5. Qualitative Evaluation


Figure 6 shows the segmentation examples produced by the PixelDiffuser for the 13 organs. In this figure, the top row represents the CT images, the second row represents the true segmentation masks (cropped), and the next 5 rows represent the segmentation (cropped) results given the number of clicks. As shown in this figure, the proposed method can discover almost the complete region of the target organ in just 3 to 5 click stages.

### 5.6. Quantitative Evaluation

As illustrated in Figure 7, we quantitatively evaluate the proposed model by comparing it with other existing models, SimpleClick [10], RITM [9], SAM [11], and FastSAM [34]. These models, like ours, can be trained without medical segmentation masks and leverage user clicks for interaction. The Dice score serves as our primary evaluation metric. For every *n* user interactions, we record the highest Dice score from the *n* segmentation masks.

Table 2 presents the average Dice score for each individual organ, while Figure 7 illustrates the average Dice score across all organs, correlated with the number of user clicks. These results indicate that our model outperforms the other methods when fewer clicks are employed. However, as the number of clicks increases, our model remains competitive with the other methodologies.

As demonstrated in Table 3, SAM-huge achieves competitive performance compared to our approach. However, SAM-huge involves a large number of parameters, leading to the necessity of substantial resources. As Table 3 presents, our model possesses significantly fewer parameters than SAM, making it computationally cost-effective and highly efficient.

Table 4 indicates a performance comparison of the approaches using the CHAOS CT dataset. As shown, our approach achieves superior performance to the other approaches, except for SAM-huge. Even though our model has about 90× fewer parameters, we can find that it achieves competitive performance with SAM-huge.

For the MRI modalities, we organize the evaluation results in Table 5 and Table 6. They indicate the DICE performance using CHAOS MRI datasets. In these cases, our method outperforms most of the comparison models; however, it is not superior to SimpleClick-huge and SAM-huge. Furthermore, even when only a single click was provided, our approach is inferior to SAM-huge. We analyze that while they are pretrained with a segmentation task, our model does not rely on the segmentation task. Using information from the segmentation task, SimpleClick and SAM are likely better at capturing the structure of objects within images compared to our approach. Nevertheless, our model contains significantly fewer parameters than SAM-huge and, considering its reduced parameter count, still demonstrates commendable performance.

### 5.7. Limitations

We showcased that our approach achieves high performance with fewer interactions compared to the previous works. In this section, we discuss some limitations involved in our method.

#### 5.7.1. Failure Cases Analysis

We observed that certain organs, such as the stomach, portal vein, splenic vein, and adrenal gland, exhibit low Dice scores, as depicted in the top left of Figure 8. Our analysis identified the primary causes for these failures: (1) The target organ appears small because it comprises only the head or tail portion of another organ. (2) The target organ is surrounded by another organ with a similar color. In such scenarios, we found that other methods produce more reliable results than our approach, as illustrated in Figure 8. We believe this discrepancy arises because our approach does not utilize any segmentation information during pre-training. This can make capturing the organ structures more challenging compared to other methods which undergo training with an image segmentation task.

#### 5.7.2. Inference Time Analysis

While performance is undoubtedly crucial in interactive medical segmentation, the inference speed at which segmentation results are produced, especially after receiving user interaction, is also important. Table 7 shows the evaluation of the inference time. As it indicates, we found that our approach takes longer time compared to other methods, mainly due to our iterative process. Although our iterative approach can yield more accurate segmentation results with fewer interactions, it requires more time to compute the masks. In our future work, reducing inference time without compromising segmentation accuracy will be our primary objective.

## 6. Conclusions

In this research paper, we introduce PixelDiffuser, a novel interactive medical segmentation approach that leverages a pretrained autoencoder, thereby eliminating the need for segmentation data. Compared to other methods [9,10,11], our proposed model can predict highly accurate segmentation with just a few clicks, a process made simple by merely clicking on the target organ area. Additionally, since our model is composed of the VGG19 encoder and the convolutional decoder, it occupies less memory than previous works which utilize huge transformer architectures as a backbone. On the other hand, we found that our method takes a long time to calculate the segmentation due to its iterative process. To speed up the inference time and to solve the failure cases, described in Section 5.7.1, will be our future work.

## Figures and Tables

**Figure 1 bioengineering-10-01280-f001:**
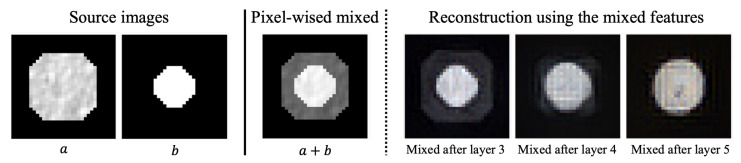
Illustration of reconstruction noises.

**Figure 2 bioengineering-10-01280-f002:**
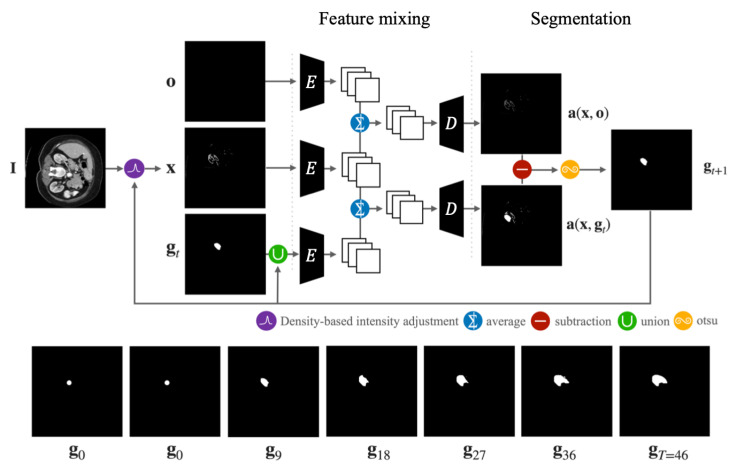
An overall illustration of our initial segmentation process. The encoder and decoder derive from the VGG19 autoencoder architecture and are pre-trained on the COCO dataset [29].

**Figure 3 bioengineering-10-01280-f003:**
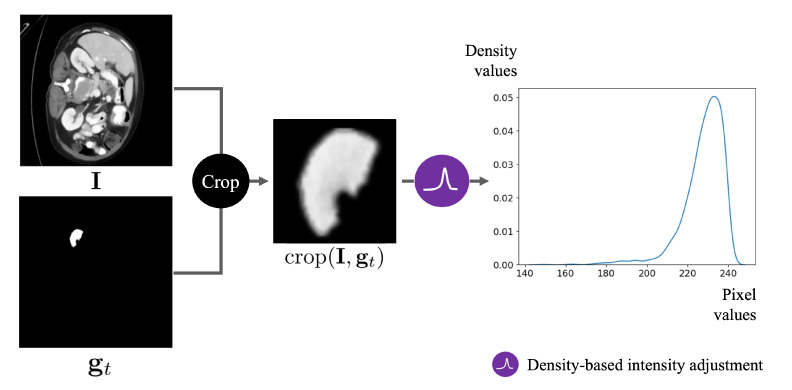
An example of kernel density estimation used to adjust the intensity values of the CT image. First, the masked region is cropped using the mask. Then, the density of pixel intensities within this cropped area is computed using the kernel density estimator. In computing the density, we exclude pixels whose value is zero.

**Figure 4 bioengineering-10-01280-f004:**
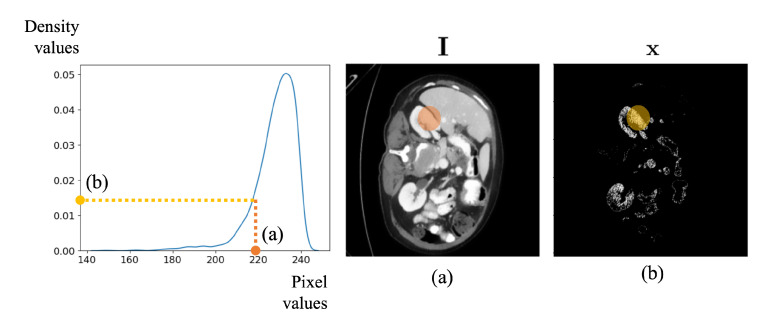
An example of the density-based intensity adjustment process. Given the density, the pixel intensities in the CT image are transformed based on the density values corresponding to each pixel value. (**a**) in this figure represents the CT image before transformation, and (**b**) is the transformed image using the density values.

**Figure 5 bioengineering-10-01280-f005:**
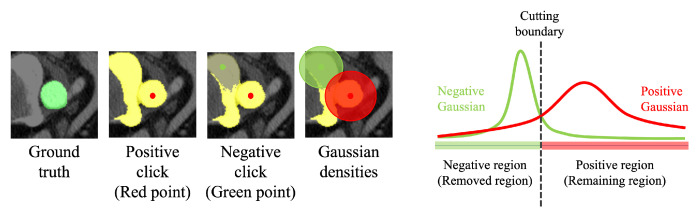
An illustration of our refinement process. The two leftmost images depict the ground truth segmentation and an initial segmentation prediction, respectively. With the initial segmentation, a user can provide additional clicks to refine the segmentation mask or choose to finalize it. If a positive click is given, our method appends an additional area to the initial mask. Conversely, a negative click discards parts of certain mask regions. The third image showcases the utilization of a negative click, where parts of the mask region are discarded when the negative Gaussian density surpasses the positive density, as demonstrated in the fourth and rightmost images.

**Figure 6 bioengineering-10-01280-f006:**
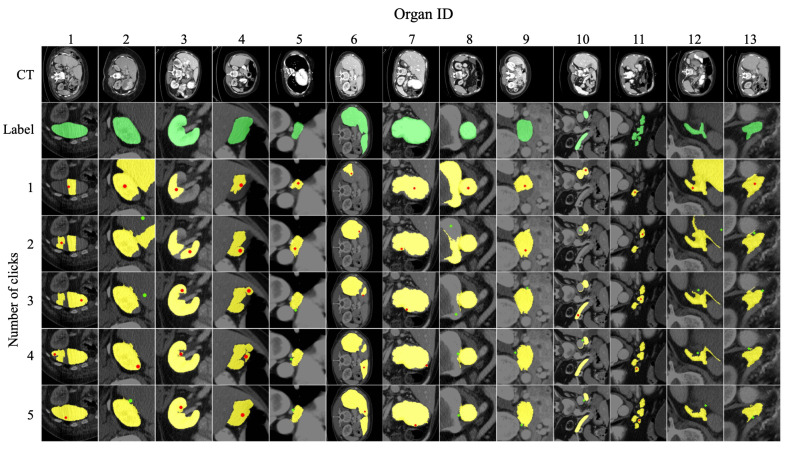
Examples of interactive segmentation for the 13 organs, varying by the number of clicks.

**Figure 7 bioengineering-10-01280-f007:**
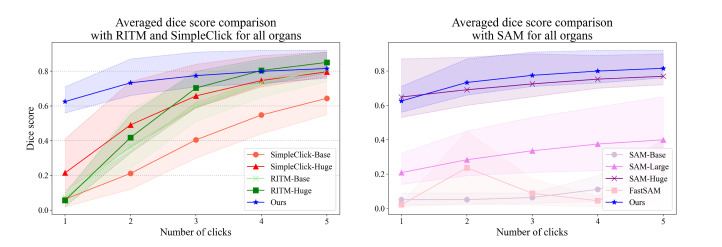
Quantitative evaluation. The solid lines represent the average dice score of the 13 organs, while dashed lines represent the 25 and 75 quantile lines.

**Figure 8 bioengineering-10-01280-f008:**
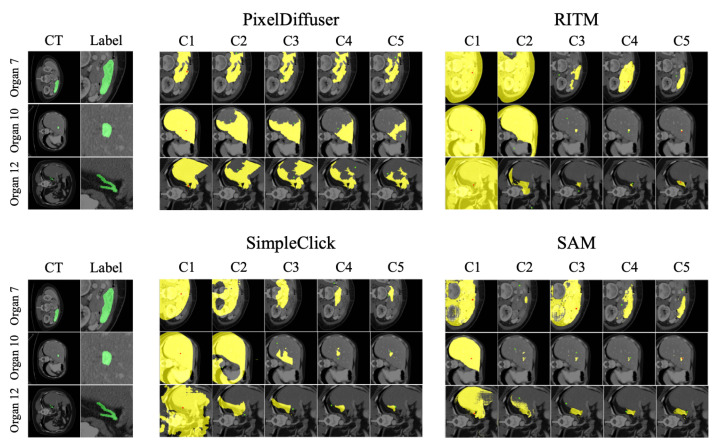
An illustration of failure cases of our approaches, and results of other methods. When a target organ is too small or surrounded by another organ with a similar texture, our method tends to fail to capture a correct segmentation mask. This is a limitation of label-free texture-based segmentation methods, involving our approach.

**Table 1 bioengineering-10-01280-t001:** The autoencoder layers of PixelDiffuser. For each Conv layer, parameters are listed in order: number of input channels, number of output channels, kernel size, stride, and padding, respectively. The MaxPool layer parameters, listed, respectively, are kernel size and stride. The Upsampling layer is parameterized by its scale factor. Additionally, all Conv layers involve subsequent ReLU activation functions.

Encoder	Decoder
Conv 0 (3, 3, 1, 1, 0) Conv 1_1 (3, 64, 3, 1, 0) Conv 1_2 (64, 64, 3, 1, 0) MaxPool (2, 2)	Conv 3_1 (256, 128, 3, 1, 0) Upsampling (2)
Conv 2_1 (64, 128, 3, 1, 0) Conv 2_2 (128, 128, 3, 1, 0) MaxPool (2, 2)	Conv 2_2 (128, 128, 3, 1, 0) Conv 2_1 (128, 64, 3, 1, 0) Upsampling (2)
Conv 3_1 (128, 256, 3, 1, 0)	Conv 1_2 (64, 64, 3, 1, 0)

**Table 2 bioengineering-10-01280-t002:** Dice score evaluation for the 13 organs using BTCV dataset.

	Spleen					Right Kidney					Left Kidney				
	**Dice Score per Click**														
**Methods**	**1**	**2**	**3**	**4**	**5**	**1**	**2**	**3**	**4**	**5**	**1**	**2**	**3**	**4**	**5**
RITM-HRNet-18	0.09	0.57	0.82	0.90	0.93	0.09	0.46	0.76	0.86	0.90	0.09	0.68	0.84	0.89	0.91
RITM-HRNet-32	0.10	0.59	0.85	0.91	0.94	0.06	0.43	0.80	0.89	0.92	0.05	0.49	0.78	0.87	0.91
SimpleClick-Base	0.08	0.33	0.58	0.75	0.84	0.07	0.41	0.64	0.77	0.84	0.06	0.34	0.62	0.77	0.83
SimpleClick-Huge	0.15	0.60	0.82	0.89	0.91	0.54	0.78	0.86	0.89	0.91	0.51	0.80	0.87	0.89	0.91
SAM-base	0.09	0.09	0.09	0.11	0.15	0.05	0.05	0.07	0.19	0.48	0.05	0.05	0.05	0.08	0.16
SAM-Large	0.30	0.45	0.56	0.64	0.67	0.29	0.44	0.52	0.58	0.64	0.32	0.45	0.53	0.59	0.65
SAM-Huge	0.88	0.91	0.91	0.92	0.92	0.87	0.88	0.86	0.84	0.86	0.87	0.87	0.86	0.84	0.86
FastSAM	0.03	0.45	0.10	0.04	0.41	0.02	0.60	0.19	0.05	0.57	0.02	0.66	0.15	0.02	0.62
Ours	0.81	0.91	0.93	0.94	0.94	0.65	0.81	0.87	0.94	0.94	0.68	0.84	0.89	0.91	0.92
	**Gallbladder**					**Esophagus**					**Liver**				
	**Dice Score per Click**														
**Methods**	**1**	**2**	**3**	**4**	**5**	**1**	**2**	**3**	**4**	**5**	**1**	**2**	**3**	**4**	**5**
RITM-HRNet-18	0.04	0.24	0.54	0.73	0.82	0.08	0.44	0.66	0.77	0.83	0.21	0.32	0.56	0.74	0.84
RITM-HRNet-32	0.03	0.43	0.76	0.84	0.88	0.03	0.55	0.74	0.82	0.86	0.22	0.42	0.73	0.85	0.89
SimpleClick-Base	0.02	0.11	0.39	0.56	0.66	0.04	0.15	0.33	0.48	0.59	0.21	0.31	0.49	0.66	0.78
SimpleClick-Huge	0.18	0.47	0.65	0.76	0.81	0.09	0.36	0.59	0.71	0.77	0.27	0.55	0.77	0.87	0.91
SAM-base	0.02	0.02	0.02	0.03	0.07	0.02	0.02	0.05	0.14	0.25	0.23	0.23	0.26	0.28	0.40
SAM-Large	0.12	0.18	0.23	0.27	0.31	0.18	0.20	0.21	0.21	0.22	0.42	0.52	0.58	0.63	0.66
SAM-Huge	0.62	0.66	0.70	0.74	0.76	0.49	0.57	0.64	0.70	0.73	0.84	0.88	0.90	0.89	0.90
FastSAM	0.01	0.04	0.05	0.02	0.05	0.00	0.01	0.00	0.00	0.01	0.12	0.24	0.17	0.09	0.27
Ours	0.71	0.79	0.81	0.83	0.84	0.56	0.66	0.71	0.73	0.76	0.70	0.87	0.92	0.94	0.94
	**Stomach**					**Aorta**					**Inferior Vena Cava**				
	**Dice Score per Click**														
**Methods**	**1**	**2**	**3**	**4**	**5**	**1**	**2**	**3**	**4**	**5**	**1**	**2**	**3**	**4**	**5**
RITM-HRNet-18	0.16	0.58	0.77	0.85	0.88	0.06	0.59	0.78	0.85	0.89	0.06	0.30	0.57	0.74	0.83
RITM-HRNet-32	0.13	0.55	0.76	0.85	0.89	0.05	0.64	0.83	0.87	0.90	0.02	0.39	0.73	0.84	0.88
SimpleClick-Base	0.15	0.43	0.63	0.75	0.82	0.10	0.26	0.51	0.68	0.77	0.02	0.12	0.33	0.52	0.65
SimpleClick-Huge	0.41	0.74	0.84	0.89	0.91	0.23	0.62	0.79	0.85	0.88	0.11	0.46	0.69	0.80	0.85
SAM-base	0.10	0.10	0.12	0.18	0.29	0.03	0.03	0.06	0.21	0.42	0.02	0.02	0.03	0.08	0.19
SAM-Large	0.37	0.46	0.49	0.52	0.54	0.24	0.34	0.47	0.57	0.60	0.14	0.20	0.25	0.30	0.33
SAM-Huge	0.67	0.73	0.80	0.84	0.86	0.90	0.90	0.92	0.92	0.90	0.68	0.72	0.75	0.79	0.81
FastSAM	0.04	0.34	0.09	0.23	0.16	0.01	0.43	0.20	0.03	0.40	0.01	0.15	0.13	0.01	0.15
Ours	0.46	0.61	0.68	0.73	0.76	0.87	0.91	0.91	0.92	0.92	0.64	0.71	0.75	0.78	0.81
	**Portal Vein and Splenic Vein**					**Pancreas**					**Right Adrenal Gland**				
	**Dice Score per Click**														
**Methods**	**1**	**2**	**3**	**4**	**5**	**1**	**2**	**3**	**4**	**5**	**1**	**2**	**3**	**4**	**5**
RITM-HRNet-18	0.04	0.23	0.46	0.62	0.71	0.03	0.27	0.51	0.65	0.74	0.00	0.03	0.21	0.44	0.58
RITM-HRNet-32	0.02	0.33	0.59	0.71	0.77	0.03	0.27	0.57	0.72	0.79	0.00	0.17	0.50	0.64	0.71
SimpleClick-Base	0.03	0.12	0.28	0.41	0.52	0.04	0.13	0.30	0.44	0.55	0.00	0.02	0.08	0.16	0.25
SimpleClick-Huge	0.11	0.35	0.53	0.63	0.70	0.17	0.48	0.64	0.73	0.78	0.01	0.06	0.22	0.37	0.49
SAM-base	0.02	0.02	0.04	0.09	0.15	0.03	0.03	0.03	0.04	0.08	0.00	0.00	0.00	0.00	0.02
SAM-Large	0.15	0.19	0.21	0.22	0.21	0.15	0.20	0.24	0.27	0.29	0.01	0.01	0.02	0.02	0.03
SAM-Huge	0.59	0.62	0.65	0.69	0.71	0.53	0.60	0.65	0.70	0.72	0.14	0.22	0.30	0.37	0.41
FastSAM	0.01	0.13	0.04	0.07	0.08	0.01	0.03	0.02	0.02	0.02	0.00	0.00	0.00	0.00	0.00
Ours	0.60	0.70	0.74	0.76	0.78	0.56	0.69	0.75	0.78	0.80	0.33	0.40	0.45	0.49	0.53
	**Left Adrenal Gland **														
	**Dice Score per Click**														
**Methods**	**1**	**2**	**3**	**4**	**5**										
RITM-HRNet-18	0.00	0.08	0.31	0.52	0.65										
RITM-HRNet-32	0.00	0.17	0.50	0.64	0.72										
SimpleClick-Base	0.00	0.02	0.08	0.17	0.27										
SimpleClick-Huge	0.01	0.11	0.28	0.42	0.52										
SAM-base	0.00	0.00	0.00	0.01	0.02										
SAM-Large	0.02	0.04	0.05	0.05	0.04										
SAM-Huge	0.35	0.42	0.48	0.54	0.56										
FastSAM	0.00	0.00	0.00	0.00	0.00										
Ours	0.56	0.63	0.66	0.68	0.69										

**Table 3 bioengineering-10-01280-t003:** The number of parameters of SimpleClick, SAM and PixelDifuser. SC-huge and SC-base represent SimpleClick-huge and SimpleClick-base, respectively.

	SC-Huge	SC-Base	SAM-Huge	SAM-Base	FastSAM	Ours
params	662,184,793	98,025,513	641,090,608	93,735,472	72,234,149	7,010,959

**Table 4 bioengineering-10-01280-t004:** Dice score evaluation for the liver using CHAOS CT dataset.

Liver					
	Dice score per click				
Methods	1	2	3	4	5
RITM-HRNet-18	0.23	0.35	0.57	0.76	0.85
RITM-HRNet-32	0.25	0.48	0.77	0.88	0.91
SimpleClick-Base	0.24	0.33	0.51	0.68	0.79
SimpleClick-Huge	0.28	0.56	0.79	0.89	0.93
SAM-base	0.26	0.27	0.30	0.38	0.52
SAM-Large	0.51	0.62	0.68	0.71	0.74
SAM-Huge	0.80	0.89	0.91	0.90	0.91
FastSAM	0.14	0.45	0.12	0.14	0.44
Ours	0.73	0.89	0.93	0.95	0.96

**Table 5 bioengineering-10-01280-t005:** Dice score evaluation for the 4 organs using CHAOS T1-DUAL (MRI) dataset.

	Liver					Right Kidney				
	**Dice Score per Click**									
**Methods**	**1**	**2**	**3**	**4**	**5**	**1**	**2**	**3**	**4**	**5**
RITM-HRNet-18	0.22	0.45	0.71	0.83	0.88	0.05	0.37	0.63	0.74	0.81
RITM-HRNet-32	0.22	0.38	0.70	0.84	0.90	0.05	0.20	0.60	0.75	0.82
SimpleClick-Base	0.23	0.30	0.50	0.70	0.81	0.05	0.28	0.58	0.74	0.83
SimpleClick-Huge	0.31	0.74	0.87	0.91	0.93	0.12	0.66	0.81	0.87	0.90
SAM-base	0.23	0.23	0.24	0.27	0.36	0.05	0.06	0.06	0.06	0.08
SAM-Large	0.38	0.44	0.47	0.48	0.50	0.17	0.21	0.25	0.27	0.30
SAM-Huge	0.74	0.82	0.86	0.86	0.88	0.78	0.82	0.85	0.85	0.86
FastSAM	0.09	0.14	0.09	0.05	0.17	0.02	0.12	0.02	0.03	0.10
Ours	0.42	0.60	0.71	0.77	0.81	0.56	0.69	0.76	0.79	0.82
	**Left Kidney**					**Spleen**				
	**Dice Score per Click**									
**Methods**	**1**	**2**	**3**	**4**	**5**	**1**	**2**	**3**	**4**	**5**
RITM-HRNet-18	0.05	0.24	0.55	0.70	0.79	0.07	0.35	0.66	0.80	0.86
RITM-HRNet-32	0.05	0.22	0.61	0.75	0.82	0.07	0.39	0.72	0.84	0.89
SimpleClick-Base	0.06	0.31	0.67	0.79	0.85	0.08	0.27	0.60	0.75	0.83
SimpleClick-Huge	0.17	0.61	0.81	0.87	0.90	0.13	0.56	0.78	0.86	0.90
SAM-base	0.05	0.06	0.06	0.09	0.16	0.08	0.08	0.08	0.10	0.14
SAM-Large	0.24	0.27	0.33	0.37	0.36	0.20	0.29	0.33	0.38	0.37
SAM-Huge	0.76	0.82	0.84	0.86	0.86	0.76	0.78	0.82	0.84	0.85
FastSAM	0.02	0.16	0.04	0.07	0.11	0.03	0.18	0.02	0.11	0.10
Ours	0.56	0.70	0.75	0.78	0.81	0.51	0.69	0.77	0.81	0.84

**Table 6 bioengineering-10-01280-t006:** Dice score evaluation for the 4 organs using CHAOS T2-SPIR (MRI) dataset.

	Liver					Right Kidney				
	**Dice Score per Click**									
**Methods**	**1**	**2**	**3**	**4**	**5**	**1**	**2**	**3**	**4**	**5**
RITM-HRNet-18	0.23	0.44	0.69	0.82	0.88	0.06	0.57	0.78	0.85	0.89
RITM-HRNet-32	0.24	0.42	0.71	0.84	0.89	0.06	0.42	0.78	0.85	0.89
SimpleClick-Base	0.28	0.38	0.61	0.76	0.83	0.21	0.41	0.64	0.75	0.83
SimpleClick-Huge	0.33	0.61	0.83	0.90	0.92	0.63	0.83	0.89	0.92	0.93
SAM-base	0.28	0.29	0.30	0.33	0.39	0.08	0.08	0.10	0.14	0.22
SAM-Large	0.51	0.57	0.62	0.65	0.67	0.54	0.62	0.67	0.73	0.74
SAM-Huge	0.64	0.76	0.82	0.85	0.87	0.80	0.86	0.86	0.88	0.89
FastSAM	0.12	0.28	0.12	0.09	0.31	0.02	0.67	0.14	0.09	0.56
Ours	0.45	0.63	0.74	0.79	0.83	0.79	0.87	0.90	0.91	0.91
	**Left Kidney**					**Spleen**				
	**Dice Score per Click**									
**Methods**	**1**	**2**	**3**	**4**	**5**	**1**	**2**	**3**	**4**	**5**
RITM-HRNet-18	0.05	0.43	0.72	0.83	0.88	0.08	0.54	0.80	0.87	0.91
RITM-HRNet-32	0.05	0.46	0.76	0.85	0.88	0.09	0.66	0.84	0.89	0.91
SimpleClick-Base	0.33	0.56	0.71	0.80	0.86	0.39	0.64	0.79	0.86	0.90
SimpleClick-Huge	0.66	0.83	0.88	0.90	0.92	0.64	0.83	0.88	0.92	0.93
SAM-base	0.07	0.08	0.09	0.13	0.24	0.11	0.11	0.13	0.16	0.23
SAM-Large	0.55	0.63	0.70	0.76	0.76	0.41	0.52	0.64	0.68	0.69
SAM-Huge	0.78	0.82	0.86	0.87	0.87	0.86	0.89	0.90	0.90	0.92
FastSAM	0.02	0.73	0.12	0.08	0.59	0.04	0.65	0.09	0.06	0.55
Ours	0.79	0.88	0.90	0.91	0.91	0.70	0.84	0.88	0.90	0.91

**Table 7 bioengineering-10-01280-t007:** Inference time comparison. For each method, we measure the inference time for each organ and calculate the min, max, and mean values. In these results, we can find the difference between the min and max time of the PixelDiffuser.

Infer Time (s)	RITM	SimpleClick	SAM	FastSAM	PixelDiffuser
Best	0.1134	0.7258	1.0502	0.0898	2.0513
Worst	0.1230	0.7708	1.0519	0.7536	5.6538
Mean	0.1183	0.7383	1.0508	0.1438	3.2868

## Data Availability

All datasets used in this research are publicly available online. The BTCV dataset can be accessed from https://www.synapse.org/#!Synapse:syn3193805/wiki/217789 (accessed on 6 January 2023), and the CHAOS dataset can be downloaded from https://chaos.grand-challenge.org/ (accessed on 9 October 2023).

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
