# Peer review of "Pixel Diffuser: Practical Interactive Medical Image Segmentation without Ground Truth"

_bioengineering, 2023, doi:10.3390/bioengineering10111280_

Round 1

Reviewer 1 Report (Previous Reviewer 1)

Comments and Suggestions for Authors

The paper: "Pixel Diffuser: Practical Interactive Medical Image Segmentation

without Ground Truth" describes a method for performing the segmentation of organs in the axial view of computed tomography (CT) images. The segmentation method is interactive and does not require ground truth images. The segmentation method gradually obtains the final segmentation by modifying a region of interest (ROI) selected by a mouse click. The method uses a pre-trained VGG19 neural network as an autoencoder and an additional neural network as a decoder.

The authors have incorporated the required corrections, and the quality of the paper has been improved. There are, however, two issues that must be addressed before publication:

  1. Utilization of the Otsu thresholding method should be justified. The Otsu method is unreliable when there are intensity gradients in the image, and the authors should test other thresholding methods more robust to spatial gradients in the image.

  2. The results section has several tables with columns labeled C1, C2…C5. However, the meaning of these labels is not defined. The authors must clearly define these labels.

Comments on the Quality of English Language

 Minor editing of English language required

Author Response

Please refer to attachment

Reviewer 2 Report (Previous Reviewer 2)

Comments and Suggestions for Authors

The authors have addressed the proposed issues. Some related papers could be cited:

Zhou, L., Wang, S., Sun, K., Zhou, T., Yan, F., & Shen, D. (2022). Three-dimensional affinity learning based multi-branch ensemble network for breast tumor segmentation in MRI. Pattern Recognition129, 108723.

Comments on the Quality of English Language

The authors have addressed the proposed issues.

Author Response

Please refer to attachment.

Reviewer 3 Report (Previous Reviewer 3)

Comments and Suggestions for Authors

All comments have been resolved.

Author Response

Please refer to attachment.

This manuscript is a resubmission of an earlier submission. The following is a list of the peer review reports and author responses from that submission.

Round 1

Reviewer 1 Report

Comments and Suggestions for Authors

The paper: "Pixel Diffuser: Practical Interactive Medical Image Segmentation

without Ground Truth" describes a method for performing the segmentation of organs in the axial view of computed tomography (CT) images. The segmentation method is interactive and does not require ground truth images. The segmentation method gradually obtains the final segmentation by modifying a region of interest (ROI) selected by a mouse click. The method uses a pre-trained VGG19 neural network as an autoencoder and an additional neural network as a decoder.

Although the topic is interesting and good results are obtained during the validation phase of this method, the paper has several issues that must be corrected to improve the quality of this document. The list of issues is presented below:

  1. The abstract section must be improved by clearly describing the methodological basis of the segmentation method. The authors must also mention how this tool was validated and the medical imaging modality used.

  2.  The discussion in the introduction needs to be more profound. The authors should discuss the state of the art concerning general medical imaging segmentation methods and their limitations. They should briefly discuss the imaging modalities that are the application target and the actual clinical application.

  3. The description of the methodology must be improved. In Figures 3 and 4, it needs to be clarified what is the Kernel density estimator. The authors should provide the equation and a reference. The Size of the crop image should be specified. It needs to be made clear what parts of the VGG19 are used and what parts are modified. The authors should elaborate on this topic. In Figure 2, the authors should indicate the blocks corresponding to the VGG19 Neural Network. The authors mention an additional neural network as a decoder but do not show this network in Figure 2. The architecture description of the neural network needs to be included. 

  4. Concerning the validation, the authors mention an extensive dataset of images used for validation. However, It needs to be clarified how the ground-truth images used to validate the method were obtained.

  5. In section 5.7, considering the failure case analysis, the authors should compare it to the other programs. In addition, the authors should discuss the possible solutions or improvements necessary for solving these failures. 

  6. The authors should include examples of other structures, such as the femur bones (or the ribs) in the sagittal (cranial) view or the left ventricle myocardium in the axial view.

  7. The conclusion section should be improved. The authors must elaborate on their approach's advantages, limitations, and future research necessary to improve this tool.

  8.  
  1.  

Comments on the Quality of English Language

Moderate editing of English language required

Reviewer 2 Report

Comments and Suggestions for Authors

The following issues should be addressed before it can be considered for publication.

 1. How is the process of Feature mixing defined? Averaged element-wise is it a simple average or a weighted average?(section4.1.2, line 157)

2. Does the size of the crop mask region in Figure 3 have any effect on the performance of the method?(section 4.1.1)

3. How well does the method work on other medical image datasets? Is the performance robust on medical images of other modalities?(section 5.1, line 187)

4. What model of GPU was used for the experiments in this paper?(section 5.3, line 210)

5. Figure 5 defines the red point as a negative click and the green point as a positive click. What do the red point and green point in Figures 6 and 8 represent?(fig.5 -4.2  fig.6 -5.2  fig.8-5.7)

6. Does the method proposed in the paper have an advantage over variant SAM methods such as FastSAM in terms of evaluation metrics such as DICE, number of parameters and inference time? (section 5.6  line 250)

7. In Interactive image segmentation methods, especially SAM and other fine-tuning models, inference speed is also an important evaluation metric, does the method in this paper have an advantage in inference speed?(section 5.6 line 255)

Comments on the Quality of English Language

The writting should be checked carefully.

Reviewer 3 Report

Comments and Suggestions for Authors

This paper presented a autoencoder based interactive medical image segmentation method. Experiment results show that it is sueprior to RITM, SimpleClick and SAM.

Comments:

1. What kind of segmentation is involved in this study? Is it semantic, instance, or panoptic?

2. The KDE function plays an important role in the expansion of original mask. However, how it is determined is not stated.

3. How long will it take to segment in practice?

4. Although MedSAM is trained on a large-scale medical image segemntation data, it can be applied in this case by transfer learning or finetuning. If so, what the performance will be?
